Low relative error in consumer-grade GPS units make them ideal for measuring small-scale animal movement patterns

Breed Greg A. 1 3 gabreed@alaska.edu
Severns Paul M. 1 2
1 Harvard Forest, Harvard University , Petersham, MA , United States of America
2 Department of Botany and Plant Pathology, Oregon State University , Corvallis, OR , United States of America
3 Current affiliation: Institute of Arctic Biology, University of Alaska , Fairbanks, AK , United States of America
Roberts David
Electronic publication date: 2015 Aug 20
Publication date: 2015
Volume: 3
Electronic Location ID: e1205
Received 2015 Jun 3; Accepted 2015 Jul 31
Copyright: © 2015 Breed and Severns
Copyright year: 2015
Copyright holder: Breed and Severns
License: This is an open access article distributed under the terms of the Creative Commons Attribution License, which permits unrestricted use, distribution, reproduction and adaptation in any medium and for any purpose provided that it is properly attributed. For attribution, the original author(s), title, publication source (PeerJ) and either DOI or URL of the article must be cited.
License URL: https://creativecommons.org/licenses/by/4.0/

Keywords: Tracking methods, Euphydrays, Butterfly movement, Animal tracking, Insect movement, Movement ecology, Checkerspot butterflies

Funding: NSERC SERDP Funding was provided by an NSERC Banting postdoctoral fellowship to GAB and partial support was provided by SERDP (Strategic Environmental Research and Development Program). The funders had no role in study design, data collection and analysis, decision to publish, or preparation of the manuscript.

==============================
Consumer-grade GPS units are a staple of modern field ecology, but the relatively large error radii reported by manufacturers (up to 10 m) ostensibly precludes their utility in measuring fine-scale movement of small animals such as insects. Here we demonstrate that for data collected at fine spatio-temporal scales, these devices can produce exceptionally accurate data on step-length and movement patterns of small animals. With an understanding of the properties of GPS error and how it arises, it is possible, using a simple field protocol, to use consumer grade GPS units to collect step-length data for the movement of small animals that introduces a median error as small as 11 cm. These small error rates were measured in controlled observations of real butterfly movement. Similar conclusions were reached using a ground-truth test track prepared with a field tape and compass and subsequently measured 20 times using the same methodology as the butterfly tracking. Median error in the ground-truth track was slightly higher than the field data, mostly between 20 and 30 cm, but even for the smallest ground-truth step (70 cm), this is still a signal-to-noise ratio of 3:1, and for steps of 3 m or more, the ratio is greater than 10:1. Such small errors relative to the movements being measured make these inexpensive units useful for measuring insect and other small animal movements on small to intermediate scales with budgets orders of magnitude lower than survey-grade units used in past studies. As an additional advantage, these units are simpler to operate, and insect or other small animal trackways can be collected more quickly than either survey-grade units or more traditional ruler/gird approaches.

Introduction

Movement ecology is developing into an important sub-discipline of ecology. This new field has developed for several reasons. First, animal tracking technology has become relatively inexpensive, light, and reliable, and tracking devices are becoming available for a broader range of species (Cagnacci et al., 2010; Block et al., 2011), with some technologies even permitting tracking of large insects (Osborne et al., 1999; Chapman, Drake & Reynolds, 2011). Second, while this technological revolution has been taking place, or perhaps because of it, population and behavioral ecologists are becoming increasingly aware of the importance of understanding movement. Movement processes are key in the maintenance of metapopulation genetic diversity and population viability, intra- and interspecific interactions, predator–prey relationships, connectivity of animal subpopulations, and they often figure prominently in conservation plans (Hanski, 1999; Dover & Settele, 2009; Knowlton & Graham, 2010; Morales et al., 2010; Costa, Breed & Robinson, 2012; Fagan et al., 2013). Along with the development of tracking technologies and increased interest in animal movement, advances in computing have allowed the development of powerful statistical models that can fit the complex time-series of animal movements to robustly address a wide range of ecological questions (e.g., Blackwell, 2003; Jonsen, Flemming & Myers, 2005; Breed et al., 2012; McClintock et al., 2013).

Although much of the new tracking data measure movement on broad spatial and temporal scales, movement measured at smaller scales is relevant to an animal’s overall behavioral repertoire, kinematics, and physiology (Goldbogen et al., 2006; Weimerskirch et al., 2006; Naito et al., 2010). In the early years of movement ecology, insects were preferred study organisms for understanding movement at these small scales. Insects are small, often abundant, and many have local movements that are easy for a human observer to follow and observe. Moreover, a careful human observer is unlikely to alter the behavior of an insect being followed, where similar strategies would be extremely disruptive if applied to vertebrates (Kareiva & Shigesada, 1983; Galen & Plowright, 1985; Haddad, 1999; Schultz & Crone, 2001; Morales, 2002). These convenient properties made insects common choices for many classic early movement ecology studies (Kareiva & Shigesada, 1983; Odendaal, Turchin & Stermitz, 1989; Turchin, 1998).

As contemporary movement ecology study systems, insects have fallen out of favor. Electronic tracking tags are now small enough to be deployed on a wide range of small mammals, birds, and fishes, and virtually all mammals larger than 10 kg. Once deployed, these tracking tags can monitor movements for weeks, months, and sometimes years virtually without effort. The relative ease and large quantity of data produced from tagging technologies, as compared to the labor intensive methods used in the classic insect movement studies, has vastly broadened the size spectrum of model taxa in movement ecology. With the exception of a few robust species (Kissling, Pattemore & Hagen, 2014), insects are not able to carry such tags and are unsuitable for such tagging protocols.

However, there remains interest in understanding how insects, and other animals too small to affix tracking tags, move and use space. If an organism’s movements are very small and constrained, they can be effectively captured with video cameras (Noldus, Spink & Tegelenbosch, 2002). For animals that mix both small (centimeter-scale) and large (meter-scale) movements together, common in many flying insects (Schultz & Crone, 2001; Severns & Breed, 2014; Breed, Severns & Edwards, 2015), video methods are usually unworkable because the animals will quickly move out of frame. There have been some successful attempts to track insects by affixing tags to them, most notably harmonic radar (Osborne et al., 1999; Ovaskainen et al., 2008; Kissling, Pattemore & Hagen, 2014). Harmonic radar, however, is only viable for the very largest, strongest fliers such as bees, dragonflies, and some butterflies. The radar systems also require line-of-sight detection and a powerful and expensive radar unit, and for many insect species harmonic radar antennae physically interfere with flight (Kissling, Pattemore & Hagen, 2014). Thus, for the majority of insects, traditional manual tracking remains the most practical approach in the field. These approaches require observers to follow individual insects, and place markers, usually pin flags, behind animals as they move. Marked locations are subsequently measured using a tape, laser range finder, pre-installed reference grid, or similar on-the-ground measurement methods (Turchin, 1998). Such methods are often extremely labor intensive, and accurate locations across the landscape are limited by the boundaries of the reference grid.

In the past 20 years, hand-held GPS units, in parallel with the aforementioned tracking technology, have greatly improved in quality and these devices are now a common piece of field gear for nearly all field ecologists. The reported error on these units usually ranges from 3 to 10 m. Researchers needing sub-meter accuracy, often required to study insect movement, have instead opted to use survey-grade GPS base units with hand-held receivers. These systems commonly cost $20,000–$40,000 USD, require dedicated training, and often proprietary software to run. Moreover, these systems are cumbersome compared to consumer grade GPS units and difficult to operate in the field. By comparison, consumer-grade GPS units sell for a few hundred dollars and promise accuracy no better than 3 m. Ostensibly, this low accuracy (absolute error, error with respect the surface of the earth) should be too coarse for insect tracking studies, but these relatively inexpensive GPS units have become considerably more accurate and precise over the last decade (Arnold & Zandbergen, 2011). More importantly, the nature of the error reported by the devices and manufacturers is often misunderstood by practicing ecologists, even those who use the technology frequently or deploy units on animals to track them. For many movement questions, however, the absolute error (which can be relatively large) is not important. Instead, the relative error, the error relative to the movement process being measured and that occurs over the time span in which consecutive points are collected, is the relevant uncertaintly. Because of the nature of GPS error, relative error can be very small on short time scales (seconds to a few minutes). Thus consumer grade GPS units can provide extremely faithful data on animal movement, even when movements are very small.

Here, we demonstrate the high accuracy of consumer grade GPS units for the measurement of insect movement by presenting and analyzing data from two grassland associated checkerspot butterflies, Euphydryas phaeton and E. editha taylori. We additionally perform a ground-truth experiment by setting up a simulated movement pathway of a scale similar to butterfly movement, with step-lengths ranging from 70 cm to 12.1 m as measured by a field tape. We discuss how to achieve the degree of GPS accuracy suitable for insect movement studies, and the counter-intuitive data collection method which favors the use of the inexpensive, consumer grade GPS over the considerably more expensive professional survey-grade units.

Methods

As part of a larger study on butterfly movement and ecology, we used consumer-grade GPS units to track two species of checkerspot butterfly in two geographically distinct regions: Euphydryas phaeton (Baltimore checkerspot) and Euphydryas editha taylori (Taylor’s checkerspot). We used consumer grade GPS units in this instance because survey grade units were not available. Both species occupy open, graminoid-dominated habitats, one in eastern (E. phaeton) and the other in western (E. e. taylori) North America. Populations of these two butterflies have relatively recently, within the last 30–120 years, adopted Plantago lanceolata (English plantain), an exotic species to North America, as the primary larval host plant.

Butterfly tracking and GPS accuracy

We tracked female checkerspots at a total of four sites, two in Massachusetts for E. phaeton (the Bullit Reservation, near Asheville, MA and Stevens-Coolidge Place, North Andover, MA) and E. e. taylori was tracked on two privately owned sites near Corvallis, Oregon (see Severns & Breed, 2014 for details). Butterflies were tracked through open grasslands and wetlands of moderate scale (∼0.8 to 20 ha) that were bordered by closed-canopy forest matrix.

Female checkerspots were selected opportunistically and tracking of selected individuals began immediately. Butterflies were followed, usually keeping a distance of 1–2 m and GPS locations were recorded every 15 s for 15 min for E. phaeton and every 20 s for 20 min for E. e. taylori with a consumer-grade, Garmin eTrex Venture HC. We enabled the WAAS setting (wide area augmentation system) on each unit, which enables greater accuracy through real time corrections using multiple satellite and on-the-ground reference stations (Arnold & Zandbergen, 2011).

We used two closely related methods for measuring movement tracks with GPS units. First, we used the traditional method of recording a movement path by placing pin flags marking each butterfly’s path at the prescribed interval. When butterflies remained at the same position for more than one time step, the number of intervals it remained at that location was recorded (Turchin, 1998). When the observation period ended we revisited each pin flag marking the animal’s path and recorded a GPS waypoint at each flag in the order the flags were placed. This required two trips around the trackway, one to place the flags while the butterfly was moving and another to take a waypoint at each flag some time later.

Alternatively, waypoints were collected in real time, at each time step while the individual butterfly was being tracked. When individuals were alighted this was done by positioning the GPS unit above the individual at a distance that did not affect behavior, usually about a meter. While in flight, positions were taken directly behind the individual, along the flight path it had just made. Except for the lack of pin flags, the positions themselves were collected the same way, by taking an individual waypoint (the tracklog function was not used). Waypoints were not automatically collected, and instead were prompted by a recorded beep that was programmed to sound every 15 or 20 s from an mp3 player. Because pin flags were not used the course does not need to be revisited twice, which is less labor intensive and caused less trampling damage to habitat as compared to the pin flag approach. In practice, collecting waypoints while tracking butterflies did not cause any more abnormal behavior than the more traditional pin flag method, though care must be taken not to hold the GPS unit such that it casts a shadow on the tracked butterfly. The interval between consecutive GPS locations is thus 15 s, slightly longer than using the pin flag method. Most of the data presented here was collected using this second methodology.

In many cases we were able to follow individuals for the entire prescribed observation interval of 15 or 20 min, but occasionally individuals could not be followed or were otherwise lost, so some tracks were shorter in duration than others. Flights of both species may be cut short by sudden bouts of inclement weather, abruptly causing butterflies to become completely still. To account for these issues, we employed rules for aborting behavioral observations if butterflies entered a protracted bout of resting during the observation period. If we observed 5 min of continuous inactivity we terminated the tracking effort for that individual and another was selected (either immediately if the weather permitted, or when weather improved). Tracking occurred when the weather conditions were ideal for buttery flight, in full or nearly full sun, and between 1000 h and 1700 h.

At every time step we recorded behavioral information, enabling the construction of behavioral budgets (reported in Severns & Breed, 2014; Breed, Severns & Edwards, 2015). However, the behavioral data also indicated whether the tracked individual moved or did not move during a particular time step. For our analysis, this is key, because a GPS location was collected at every time step regardless of whether the butterfly moved. This enabled us to compare actual butterfly movements to “null” movements. “Null” movements are apparent moves made according to the GPS data, but through our direct observations, we know that no movement actually took place. Thus, the null moves are strictly attributable to GPS error and represent a control. This allowed us to sort the steps represented by each consecutive GPS location into those representing movement and those representing only GPS error. We then compared the step-lengths in butterfly pathways where the individual moved (true steps) to the step-lengths where there was no movement (null steps). We used these errors to estimate the signal to noise ratio in the movement pathways and the patterns associated with the error over time (direction and magnitude).

Ground truth track

To better understand and confirm the butterfly tracking results and null vs. true comparisons, we set up a ground-truth track in the University of Alaska’s experimental forest land. The track was simply a set of numbered pin flags marking the ends of consecutive steps of known length, which we prepared using a field tape and compass to measure step-length and bearing. The course was 11 points that formed a jagged loop with mixtures of step-lengths, numerous turn-backs and sharp corners, but also more subtle turn angles; all common features of a real butterfly pathways. Step-lengths were mixed and ranged from 70 cm to 12.1 m. The course was measured 20 consecutive times, with points collected at the vertices of the track using the same protocol as that used for butterfly tracking, with a point collected every 15 s. The step-lengths and step-bearings measured from the 20 ground truth tracks could then be compared with each other for consistency and variation across the sample and also with the field tape measured step-lengths and compass measured step-bearings.

This ground truth was done in a different year than the butterfly tracks (2015) and also used a different GPS model (Garmin GPSmap 62s). However, it is clear from the data that the different units have similar properties with respect to fidelity in reproducing movement pathways and we believe the results are generally valid for Garmin units. We cannot extend our inferences to other manufacturers, but Garmin manufactures more than 50% of the consumer grade GPS units in North America and close to 20% in Europe so these units should be widely available to practicing ecologists. Moreover, the simple ground truth we present here can be easily accomplished with other GPS models and manufacturers.

Results

Null vs. true steps in butterfly tracking data

We found that with respect to the scale of real butterfly step-lengths, GPS error was very small, with the average null step-length being 0.378 m and a median null step-length of 0.11 m. 37% of null steps were 0.0 m, 80.3% of null steps were less than 0.25 m and 94.8% of null steps were less than 1 m. Four example tracks are shown in Fig. 1, and results summarized in Table 1 and Fig. 2. The relatively large mean of null step-lengths was due to a handful of very large erroneous steps. Real steps (across both species) had mean and median lengths of 3.14 m and 2.12 m, respectively. Comparing means, the signal to noise ratio was 8.4, while the signal-to-noise ratio of the median, a better estimate of central tendency, was 19.3.

Figure 1 Example butterfly tracks.

Four example butterfly tracks, the starting points are set to 0 and the scale is in meters. The red tracks show the true steps connected, while the black tracks connect the null steps for comparison.

Figure 2 Histograms of null vs. real step-lengths.

Histograms of null steps (A) and true steps (B).

Figure 3 Ground-truth trackways.

Ground-truth track: the blue line represents a fixed array of pre-measured points that were traced 20 times using the GPS tracking method described for butterflies, with locations collected every 15 s. The track itself is of approximately the same scale a butterfly might move, with step-lengths ranging from 70 cm to 12.1 m. Step-lengths were measured with a field tape while bearings were measured with a compass. Note that although there is a great deal of error around the points themselves, the consecutively collected points represent both step-length (Fig. 4) and step-bearing (Fig. 5), and by extension turning angle, accurately. Thus if the ground-truth track were actually the path of a flying insect, its movement pattern would be faithfully captured. Also note that, using a compass to measure bearing in the field has relatively large error, so the GPS average locations at each of the track’s vertices more accurately represent the ground-truth track than the field tape and compass readings, which have been left out of this figure.

Table 1 Summary of null and true step-lengths.

Comparison of null steps (butterfly was directly observed to not move) and real steps (butterfly was directly observed to move) as measured by consumer-grade GPS units.

Statistic	Null steps	Real steps	
Number (N)	2,830	1,228	
Median (m)	0.111	2.12	
Mean (m)	0.378	3.14	
Number = 0	1,042 (36.8%)	21 (1.7%)	
Number < 0.25 m	2,275 (80.3%)	144 (11.7%)	
Number < 1.00 m	2,684 (94.8%)	358 (29.1%)	
Number < 3.00 m	2,759 (97.5%)	739 (60.1%)	
Number > 10.0 m	20 (0.7%)	75 (6.1%)	

Ground-truth trackway

Results from the ground-truth trackway are shown in Fig. 3. The absolute error around each of the track’s vertices is clear, but the small relative error in the individual step-length and bearings is also clear. Steps are all nearly parallel, except for the very shortest steps which have a bit more variation in their bearing, but even in these cases the bearings are all constrained to be in the same quadrant. Median errors differed for each step, but were mostly between 20 and 30 cm, and mean errors ranged from 22 to 48 cm, which is slightly larger than the median and mean null steps from the real tracking data, but still a very favorable signal-to-noise ratio. Step-lengths were also slightly overestimated (Fig. 4).

Figure 4 Ground truth track step-length histograms.

Bins are 20 cm wide. Note that there is some error around step-lengths as there would be in other tracking technologies, but that the median error (median difference between GPS-measured step-length and tape-measured step-length) is small relative to the step size, even for steps in of 0.7 to 1.5 m. In addition, the distributions of the various step sizes overlap very little. For reference, step 1 is the displacement between points 0 and 1, step 2 between points 1 and 2, etc. Step 11 is the displacement between points 10 and 0 (see Fig. 3 for reference).

Figure 5 Ground-truth track step-bearing circular histograms.

Bins are 60° wide. Although short steps have a less favorable signal-to-noise ratio, their bearings are consistent from one track to the next, with well constrained headings that are in line with those measured by a hand-held compass in the field. In fact, for very short steps an accurate compass reading (“True Bearing”) in the field is difficult (∗ such as step 8), and bearing calculated from the GPS tracks appear to be more accurate than the field measured compass bearing.

Step-bearings were also well constrained. For short steps, the entire distribution of steps was constrained within a 45° section of a circle, for longer steps measured bearings fell within a 15° section of a circle (Fig. 5). For short steps, GPS measured bearings appeared less biased and thus more accurate than those measured in the field by a compass. The accurate step-lengths and bearings represent turning angle, a key aspect of animal movement, extremely well. Although there is still observation error in these ground-truth trackways, the movement pattern is clearly visible, even in each individual track, and if this were real data, would be highly suitable for a wide range of analytical approaches to address a number of animal movement questions.

Discussion

To understand why error is so small relative to the movement being measured, even when the moves themselves are small, compared to the larger error radii reported by GPS units, it is important to understand that these errors are fundamentally different from those reported by GPS units and the manufacturers. Error reported by GPS units is absolute error, it is measured with respect to fixed positions on the surface of the earth. Here we are interested in the error relative to animal movement step-lengths, which are recorded by two GPS locations taken in quick succession (15 or 20 s, in this case). The reason errors are so small has to do with the “slow-varying” nature of GPS error, whereby, on short time scales, error residuals are highly autocorrelated. This means that the underlying movement steps, even when very small, will faithfully represent movement because the starting and ending GPS locations are displaced from the surface of the earth with approximately the same direction and magnitude. If the error residuals of the starting and ending GPS locations are the same, the underlying movement step will be correct. This effect is clear in the ground-truth tracks plotted in Fig. 3. The slow-varying nature of GPS error and high autocorrelation of residuals is well illustrated by Arnold & Zandbergen (2011), who carefully examine the nature of error in consumer grade GPS units.

Our experiments with tracking accuracy using consumer grade, hand-held GPS units have some important implications. First, the accuracy and precision of the step-lengths was considerably better than manufacturer states in our purpose of measuring movement. Similar conclusions were reached by the US Federal Aviation Administration and National Transportation & Safety Board with respect to the waypoints themselves. In 2008, they reported the real 95% confidence intervals using the WAAS (Wide Area Augmentation System) augmented GPS units (now standard on most consumer grade GPS units) were on the order of 0.5 to 1.0 m (FAA, 2008). The nominal error for WAAS augmented GPS units is 7.6 m. However, this 7.6 m error radius is actually the mandated upper bound of the WAAS system. It is likely that, given the reliance on GPS systems of navigation and surveying, GPS manufacturers overstate error to protect themselves against liability.

In our data, error introduced to step-lengths by consecutively collected GPS points were generally <0.25 m and the signal to noise ratio was between 9:1 and 20:1, depending upon how it was calculated. Both checkerspot butterflies made moves that averaged just over three meters, with many steps that were much shorter. Compared to the scale of movement that the tracked butterflies made during flight, GPS noise had minimal (though not negligible) impact on the movement pathway. Our ability to quantify butterfly movements and to detect differences in movement patterns between and within the two checkerspot species was high, and it was extremely easy to detect even subtle difference in movement (reported elsewhere: Severns & Breed, 2014; Breed, Severns & Edwards, 2015).

There are some elements of our sampling protocol that were likely key in obtaining highly precise results. First, observation intervals were short; 15 and 20 s. These intervals are considerably shorter than the satellite drift error process that affects GPS accuracy, with autocorrelation dissipating to zero after about 15–30 min—though we note that changes in error autocorrelation can sometimes be abrupt when consecutive positions are fixed using a different constellation of GPS satellites. Second, adult butterflies were only active under warm, calm, clear, and atmospherically settled days, optimal conditions for acquiring satellites and minimizing GPS error. Furthermore, the butterflies we tracked in open meadows, so multipath error (reflected signals) was likely minimal. Both field sites (Massachusetts, Oregon), as well as the ground-truth site (Alaska) were in North America, where WAAS network coverage is 100%. These factors likely enabled the GPS units to produce more accurate locations than could be expected beneath a forest canopy or in canyons, for example.

From our results, we cannot conclusively determine the time interval at which the GPS error residuals will no longer be correlated. But given the nature of ionospheric distortion, observation protocols that use intervals up to 1 min and perhaps 2 will likely produce results similar to our 15–20 s time intervals. Longer time steps would not be recommend unless meter-scale accuracy is acceptable. Finally, as track locations are recorded in-person (as opposed to remotely through an electronic tag), we highly recommend that observers take ancillary behavioral notes, indicating if a step occurred and approximately how long that step was, in order to ground-truth the step-lengths collected. These notes were key for us, as without records of butterfly behavior from standard behavioral observations, it would not have been possible to assess how well the GPS units performed.

Although slow drift error will introduce only a small amount of error to individual steps, it will shift entire tracks in one direction and/or slightly distort whole tracks, which are collected over 15 or 20 min. However, almost all relevant behavioral information is preserved with the consumer grade GPS units, and because the absolute error with respect to the surface of the earth is still relatively small (a meter or so using the WAAS system), habitat usage patterns will be well represented. True centimeter scale accuracy, such as that produced by survey grade GPS units, would likely only be required if two conditions are in place: (1) all features in the landscape to which an animal might respond are known at centimeter-scale accuracy (such as individual flower blossoms or host plant locations) and (2) animals actually respond to resources only at these scales and not the larger scales that might be produced by habitat edges.

For insects and other animals moving on small to medium scales in the field which can be followed by an observer without affecting behavior (e.g., Potts & Lewis, 2014; Potts et al., 2014), consumer grade GPS units offer a remarkably inexpensive and easy way to collect movement data with sub-meter accuracy and movement studies can be conducted with little up-front cost. It is important to understand how GPS error operates on step-length data, and the errors we report are with respect to the moving animal and are thus relative error, and do not represent the absolute error in position with respect to the surface of the earth, which may be much larger. Precision mapping of actual geographic locations and the boundaries of landscape elements (sub-meter accuracy) may be more labor intensive than recording movement tracks and care must be taken if precise geographic position is required. If accurately measuring step-lengths, orientations, and movement behavior of insects or other small animals though time is the study goal, as is often the case in movement ecology, consumer grade GPS units are a viable alternative to much more expensive, survey-grade, base units and rovers.

Supplemental Information

Supplemental Information 1 Raw butterfly movement data

Click here for additional data file.

We thank Ed Easterling for access to butterfly populations in Oregon, the Trustees of Reservations for population access in Massachusetts, and Vivian Kimball for assistance in the field.

Additional Information and Declarations

Competing Interests

Author Contributions

The authors declare there are no competing interests.

Greg A. Breed conceived and designed the experiments, performed the experiments, analyzed the data, contributed reagents/materials/analysis tools, wrote the paper, prepared figures and/or tables, reviewed drafts of the paper.

Paul M. Severns conceived and designed the experiments, performed the experiments, contributed reagents/materials/analysis tools, wrote the paper, reviewed drafts of the paper.

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
