# Peer review of "Low relative error in consumer-grade GPS units make them ideal for measuring small-scale animal movement patterns"

_PeerJ, doi:10.7717/peerj.1205_

## Round 0.1 · original submission · Major Revisions

The first reviewer in particular raised a number of significant concerns regarding the paper and these need to be addressed before the paper can be published.

Reviewer 1 ·

Basic reporting

This paper is easy to follow and understand, it also covers much of the recent literature in the area.

The paper reviews the discovery of “perceived” high precision within of consumer grade GPS units, using the tracking of butterfly movement.

I am unable to comment on the majority of the introduction as this is a little beyond my expertise, but from a reader's perspective it reads and sets the scene well.

I do find the title misleading, especially with the word “precision” (see experimental design comment below), but also the use of units - as only one GPS unit was tested.

Experimental design

The paper cover the quality of consumer GPS, the methods are easily reproduced and well set out, but the lack of control in the experiment, means the results and finding will inherently lack vigour, the figures quoted for precision could have little bearing in on reality.

The authors have conducted experiments using consumer tracklogs, but have failed to take into account the nuances and smoothing algorithms of consumer GPS. This is could be easily accounted for by the use of control in the experiment, but without this control the results will be flawed.

There are two main issues:


1 The lack of control, means we do not know if we are really seeing a tracked butterfly or just the noise or drift from GPS signal (it will be a mixture of the two). To be acceptable for publication that the authors will need to include some control in the experiment. This control could be very simple:
o Follow the same path with a higher resolution GPS
o Follow a standardised path: if no good point ground control is available then using a predefined shape to follow (i.e. a circle path of 10m,15m,20m,...). They should also include some more natural paths shapes (for example with abrupt movements)

2 Algorithms consumer GPS units use to produce track logs
o Consumer GPS units use many algorithms and methods to "smooth" results present to the user as a “steady” point of location. Unfortunately this information is very hard to find and will be different for different units (and possibly different firmwares). Again without control in the experiment, the influence of these algorithms would be unknown . Also newer GPS unit have, many other sensor than the GPS chip (magnetometer, barometer and accelerometer), all of which "help" the unit to calculate movement as well as its location. This has two important consequences for this paper:
a. Often units will sense that it is not moving and will therefore display and record a stable point.
b. If the unit is moving it will often have knowledge of direction and speed (from sensors or previous movements), from which is can estimate where is should be from the last locality
By bringing control in the experiment, it would be possible to see the effects of these algorithms, but also would allow different units to be compared.

There is minor issue which is connected to 2 above and that is that the authors have only tested one unit (Garmin Etrec Venture HC), but assign the findings to all consumers units. As mentions above each unit with have different nuances and therefore different stepwise changes.

Validity of the findings

As there are issues with the experimental design and no controlled is in place, I can not assess the findings meaningfully. Reading the discussion and results the following point are raised:
a. The author used two methods one using tracklogs and the other using waypoints (page 6 paragraph 2). The results from the waypoint method are not discussed or presented?
a. The authors mention the “slow varying” nature of GPS which is true but not at the scale the authors are trying to achieve. The reality is that this slow movement is within the figures that the author quote (i.e. movements of 0 - 30 cm within 10 seconds - see Arnold et al 2011).

Additional comments

This is an interesting paper and the Authors are likely correct that the accuracy of the units will be higher when recording tracklogs than the 95% accuracy often quoted. The method of using stepwise and non-stepwise recording is useful and should be pursued, but with control in place. I would encourage the authors to run the same experiments with control in place this would give real world examples of the change in accuracies.

·

Basic reporting

The basic presentation of the paper is good. The English is generally of a fully acceptable standard.
The background introduction is comprehensive with regard to the application area and the nature of the problem being attacked. Referencing of material in the area of the target application is good, although I have some reservations with regard to previous work on GPS systems which I will refer to in the next section.
All figures are relevant and well presented.
There are a few minor presentational issues. These include:
1. Mixed and inappropriate mixing of conventional and italic text - see for example the caption on figure 5.
2. There are a few minor typographic errors that need addressing - see for example second sentence page 8 "Th"
There are two aspects of technological terminology that the authors do need to address. They repeatedly use the words "precision" and "accuracy" as if they are interchangeable - they are not and mean quite different things. This should be corrected.
Secondly, the paper would benefit considerably in clarity from the adoption of the terms "relative error" to describe the error in movement, and "absolute error" to describe the error in position. This usage would bring the paper more into line with the main body of work of positional accuracy.
Generally speaking it is my view that the paper is acceptable for publication in terms of this criterion with the amendments outlined above.

Experimental design

There are some issues in this area. While the paper is very interesting - and while I am not a specialist in the application area I found the paper most pleasurable to read - and it is quite likely that the results presented do indeed constitute an original contribution in this field, they do not with regards to GPS systems.
The fact that GPS is more accurate in a differential mode rather than an absolute mode has been known for some time - as indeed even a cursory search on the term "differential GPS" will reveal.
Now the authors may well feel that differential GPS, as it is conventionally implemented, is different from what they are doing. For example they have no "base station" the position of which is accurately known independently. But technically it is not different! Errors in GPS stem mainly from atmospheric conditions - the basic assumption used in differential GPS is that these errors are the same for the base station as they are for the roving station. This paper is basically exploiting exactly the same effect. The error in each reading is consistent, as the atmospheric conditions don't change appreciably in the time scale of the experiment. So basically each reading is in error by the same amount, but point to point the error cancels - resulting in more accurate RELATIVE measurements. It is the same effect that conventional differential GPS uses.
This is an issue for the paper - in that to a worker in GPS the result is not actually that "surprising" at all!
Having said this I still feel that the paper has merit and should be published - the result is an important demonstration to fellow workers in the same field as the authors of this basic fact - that use of GPS in a relative mode is more accurate than simply recording absolute positions.
The paper is also valuable in reminding us that sometimes more prosaic experimental methods can yield robust and interesting results! A good lesson for us all.
So, I feel that if the authors can address this issue and make some explanation of differential GPS and how it guides them to this way of working - and rework the title maybe to remove the "surprise" element? Then this paper would be quite acceptable and an interesting and worthwhile contribution.

Validity of the findings

This is the weakest area for me as a reviewer. As I have stated my expertise is in sensors and so GPS systems, not ecology or "animal" movement.
But in my, albeit, limited assessment the authors do seem to go to some pains to validate their data, subject it to appropriate statistical analysis and present it in a fair and reasonable fashion.
As far as I am competent to judge this section seems quite acceptable to me.

Additional comments

I think this is a most interesting paper, and I really did enjoy reading it. I was heartened that such a simple and practical approach yielded data of such high quality.
I think there are excellent practical lessons to be drawn from the work and so I believe that it should be published. But I do think that the prior existence of differential GPS needs to be acknowledged and also, for the paper to be acceptable to those specialising in the field of measurement, it would perhaps be wise to express slightly less "surprise" at the result!

---

## Round 0.2 · accepted · Accept

Thank you for making the corrections where appropriate as suggested by the reviewers. This has made the manuscript a lot clearer and therefore I am happy to accept without further review.